# Prospects of Biochar for Sustainable Agriculture and Carbon Sequestration: An Overview for Eastern Himalayas

**Jayanta Layek** [1,*], **Rumi Narzari** [1], **Samarendra Hazarika** [1], **Anup Das** [2], **Krishnappa Rangappa** [1], **Shidayaichenbi Devi** [1], **Arumugam Balusamy** [1], **Saurav Saha** [3], **Sandip Mandal** [4], **Ramkrushna Gandhiji Idapuganti** [5], **Subhash Babu** [6,*], **Burhan Uddin Choudhury** [1] and **Vinay Kumar Mishra** [2]

1   ICAR Research Complex for North East Hill Region, Umiam 793103, Meghalaya, India;
    narzarirumi@gmail.com (R.N.); samarendra.ches@gmail.com (S.H.); krishphysiology@gmail.com (K.R.);
    shidayaish@gmail.com (S.D.); enssamy@gmail.com (A.B.); burhan3i@yahoo.com (B.U.C.)
2   ICAR Research Complex for North East Hill Region, Tripura Centre, Agartala 799001, Tripura, India;
    anup_icar@yahoo.com (A.D.); vkmishra63@yahoo.com (V.K.M.)
3   ICAR Research Complex for North East Hill Region, Sikkim Centre, Gangtok 737102, Sikkim, India;
    sauravs.saha@gmail.com
4   ICAR Central Institute of Agricultural Engineering, Bhopal 462038, Madhya Pradesh, India;
    smandal2604@gmail.com
5   ICAR Central Institute for Cotton Research, Nagpur 440010, Maharashtra, India; ramkrushnagi@gmail.com
6   ICAR Indian Agricultural Research Institute, New Delhi 110012, Delhi, India
*   Correspondence: jayanta.icar@gmail.com (J.L.); subhiari@gmail.com (S.B.); Tel.: +91-91-0101-1194 (J.L.)

**Abstract:** The net arable land area is declining worldwide rapidly due to soil erosion, drought, loss of soil organic carbon, and other forms of degradation. Intense rainfall, cultivation along steep slopes, unscientific land-use changes, shifting cultivation, soil acidity, and nutrient mining in hills and mountains make agriculture unsustainable and less profitable. Hills and mountain ecosystems of the Eastern Himalayan Region (EHR) are further prone to the impact of climate change posing a serious threat to agricultural production and the environment. Increasing soil carbon reserves contributes to multiple ecosystem services, improves soil nutrient and water-holding capacities, and advances climate-resilient agriculture. Thus, carbon sequestration is increasingly becoming an important aspect of farming among researchers in the region. The EHR predominantly practices shifting cultivation that degrades the ecosystem and promotes land degradation and biodiversity loss. Leaching of exchangeable bases is highly favored due to excess rainfall which in turn creates an acidic soil accounting for >84% of the region. Application of lime to raise the soil acidity for the cultivation of crops did not get adequate acceptance among the farming community due to multiple issues such as cost involvement, non-availability in time and place, and transportation issues. The application of biochar as soil amendments is widely known to improve soil's physical, chemical, and biological properties. Biochar has also emerged as a potential candidate for long-term carbon sequestration due to its inbuilt structure and higher stability. Shift from traditional "slash and burn" culture to "slash and char" might lead to the sequestration of carbon from the atmosphere. Around 0.21 Pg of carbon (12% of the total anthropogenic carbon emissions by land-use change) can be sequestered in the soil if the traditional "slash and burnt" practice is converted to "slash and char". The objective of this review is to provide detailed information about the role of biochar in altering the soil properties for sustaining agriculture and carbon sequestration, especially for hills and mountain ecosystems.

**Keywords:** biochar; carbon sequestration; climate change; shifting cultivation; soil properties; Himalayan Region

## 1. Introduction

The Eastern Himalayan Region (EHR) of India situated in between latitudes of 26°40′–29°30′ N and longitudes of 88°5′–97°5′ E) is blessed with a tremendous number of

resources which provides large scope for growth and development in agriculture and other important sectors [1]. Meager use of agro-chemicals, rich agrobiodiversity, strong indigenous knowledge systems, etc., offers an opportunity for organic farming in the region and thus provides great scope for attaining sustainability in agriculture [1]. However, in today's modern world, climate change poses a serious threat to agriculture due to its impact [2] and has been one of the major issues to be debated among scientists, policymakers, and organizations. The continuous increase in the atmospheric concentration of greenhouse gases (GHGs) due to anthropogenic emissions can lead to a significant change in climate [3]. The adverse effects of climate change, faulty agricultural practices, environmental degradation, and lack of suitable technologies pose a serious threat to sustainable agriculture in the region. The EHR of India has 20.2 million ha (Mha) areas in the states of Sikkim, West Bengal, Assam, Arunachal Pradesh, Manipur, Mizoram, Tripura, Meghalaya, and Nagaland and Tripura favors suitable climatic conditions for the growth of diverse agricultural and horticultural crops. However, excessive and high intense rainfall in the region causes leaching of exchangeable bases from the soil leading to the development of soil acidity. A significant area of the EHR consists of the states of Arunachal Pradesh, Manipur, Mizoram, Tripura, Meghalaya, Nagaland, Sikkim, and Tripura practice "slash and burn" cultivation which is also known as "shifting or Jhum" (local) cultivation. However, the practice leads to several environmental degradations such as soil erosion, disturbances of soil and forest ecosystem, and contribution of air pollution including greenhouse gases [4] apart from reduction in soil organic matter [5] and loss of valuable biomass and nutrients. Around 90% of the soils of the region are known to be acidic [6] which causes nutrient imbalance for the uptake of plants (especially phosphorous and micro-nutrients) and thus poses severe impacts on crop production. The presence of a threshold level of organic matter in the soil is essential to maintain the equilibrium functioning of the soil's physical, chemical, and biological integrity as well as carrying out agricultural production and environmental functions of soil [7].

The scientific communities and several other policymakers have the challenging task of developing efficient strategies and methods to sustain the degrading environments through a sustainable approach. Biochar, a co-product of pyrolysis [8] is identified to be an important tool in tackling the above issues in the EHR along with several other strategies that prevailed. It is an important and popular carbon sequestration method to mitigate climate change [9]. Carbon sequestration is the process of long-term capturing and storing of carbon dioxide in order to prevent it from entering the atmosphere. Carbon sequestration is the process of removing carbon from the atmosphere and storing it in another form that cannot immediately be released. Carbon sequestration occurs both naturally and as a result of anthropogenic activities and typically refers to the storage of carbon that has the immediate potential to become carbon dioxide gas [10]. Carbon sequestration is different from the carbon storage, which is a complex method carried out using various technologies to reduce and capture carbon dioxide and permanently storing them in geological reservoirs. The added advantage of the EHR has a vast expansion of vegetation and unused waste material which provides large scope and opportunity for biochar production. The region has the potential to produce 37 Tg (1 Tg = $10^{12}$ g) of agricultural waste and out of this, if 10% is used for producing biochar, then about 13,000 and 9000 Mg of bio-oil and biogas can be produced additionally, which is equivalent to 31 TJ of energy. The application of biochar improves soil's physical, chemical, and biological properties [11,12] by acting as soil amendments [8,13–15]. The high stability of biochar in the soil compared to other organic conditioners [16] helps in extending its potential effect on the soil quality over a longer period [17]. Biochar having pH can act as a liming agent by raising soil pH upon application and helps in amelioration of soil acidity [18]. However, there is a dearth of work reported from the sloping lands of EHR region on biochar which provides ample scope for scientific communities as research priorities of biochar application on soil quality apart from carbon sequestration. This paper is to present the possibilities of biochar production, and its application in the EHR to enhance soil quality and crop improvement, as well

as to create general awareness amongst people of its potential role in mitigating climate change through carbon sequestration. This paper also emphasizes that biochar can serve as a sustainable, cost-effective, and environmentally viable alternative to chemical fertilizer in the EHR.

## 2. Importance, Properties and Benefits of Biochar

### 2.1. Biochar and Its Importance

Biochar can be defined as the carbonaceous product that is obtained when biomass is subjected to heat treatment in an oxygen-limited environment (pyrolysis) and the charred product when applied to soil as an amendment [8]. Pyrolysis is the thermal depolymerization of biomass at elevated temperatures without the participation of oxygen. The end products of pyrolysis are syngas, bio-oil, and char [19,20]. The char can be used as an energy source, and acts as a soil amendment which is called biochar. It can be produced from a wide variety of organic materials including paper mill sludge, forestry and crop residues, and poultry waste [21]. Biochar applications gaining growing interest as a sustainable technology that helps in improving the weathered and degraded soils [22]. It enhances the soil's physical (i.e., bulk density, water holding capacity, permeability, etc.), chemical (i.e., nutrient retention, nutrient availability, etc.") and biological (microbial population, earthworm, enzyme activities, etc.) characteristics which thereby improve plant growth and development [23]. Its recalcitrant nature towards microbial decomposition guarantees a long-term benefit to soil fertility [24]. Apart from this, it also improves the saturated hydraulic conductivity of the topsoil of rice fields and xylem sap which results in higher crop yields and improved response to N and NP chemical fertilizer treatments [25]. They possess a negligible number of heavy metals or toxic elements such as As, Cd, Pb, and polycyclic aromatic hydrocarbons so contamination risk is very low. They have the potential to enhance soil fertility; crop productivity [26–29]; enhance nutrient and water use efficiencies [30], and mitigate emissions of $N_2O$ [31]. The increasing level of atmospheric $CO_2$ can be mitigated by the long-term storage of C in soil. In this regard, biochar has emerged as a viable option for sequestering carbon in soil [26].

### 2.2. Properties of Biochar

The biochar properties are greatly influenced by the feedstock source and pyrolysis conditions [31,32]. In general, wood biochar has high total C; low ash content; low total N, P, K, S, Ca, Mg, Al, Na, and Cu contents; low potential cation exchange capacity (CEC); and exchangeable cations as compared with manure-based biochar. The increase in pyrolysis temperature increased the ash content, pH, and surface basicity and decreased the surface acidity [32]. In the case of fast pyrolysis, biomass is rapidly heated to 400–550 °C and the main product is bio-oil while in slow pyrolysis, the biomass is slowly heated to the desired peak temperature and the main products are biochar and syngas [33]. Some of the important physicochemical properties of biochar are higher surface area and porosity, low bulk density, higher cation exchange capacity (CEC), neutral to high pH, and higher carbon content [34]. It also contains N, P, and basic cations such asCa, Mg, and K [35] which are essential plant nutrients for crop growth and development. Pyrolysis at low temperature yields higher biochar while biochar with higher C content, large surface area, high adsorption characteristics, greater porosity, and more stable C are obtained at higher temperatures [35]. At high pyrolysis temperature (>600 °C), the functional groups are gradually lost, leaving the material with a high degree of condensation and more recalcitrant with polycyclic aromatic structure [35,36]. The stability of biochar to sustain hundreds to thousands of years in the soil is attributed to a higher proportion of aromatic structures which in turn provides higher resistance against chemical and biological decomposition [16]. The concentrations of C and N for plant-based biochar increase with an increase in pyrolysis temperature while the concentrations of C and N in mineral-rich feedstock decrease with increasing pyrolysis temperature [31]. Some of the properties of biochar from different biomass sources are presented in Table 1.

**Table 1.** Properties of biochar derived from different sources.

| Materials Used for Producing Biochar | pH | Total C (%) | Total N (%) | C: N Ratio | Ca (cmol kg $^{-1}$) | Mg (cmol kg $^{-1}$) | P (cmol kg $^{-1}$) | K (cmol kg $^{-1}$) | Cation Exchange Capacity (cmol kg $^{-1}$) |
|---|---|---|---|---|---|---|---|---|---|
| Paper mill waste 1 (waste woodchip) | 9.4 | 50.0 | 0.48 | 104 | 6.2 | 1.20 | - | 0.22 | 9.00 |
| Paper mill waste 2 (waste wood chip) | 8.2 | 52.0 | 0.31 | 168 | 11.0 | 2.60 | - | 1.00 | 18.00 |
| Green waste (grass, cotton trash and plant prunings) | 9.4 | 36.0 | 0.18 | 200 | 0.4 | 0.56 | - | 21.00 | 24.00 |
| Eucalyptus biochar | - | 82.4 | 0.57 | 145 | - | - | 1.87 | - | 4.69 |
| Cooking biochar | - | 72.9 | 0.76 | 96 | - | - | 0.42 | - | 11.19 |
| Poultry litter (450 °C) | 9.9 | 38.0 | 2.00 | 19 | - | - | 37.42 | - | 11 |
| Poultry litter (550 °C) | 13 | 33.0 | 0.85 | 39 | - | - | 5.81 | - | 11 |
| Wood biochar | 9.2 | 72.9 | 0.76 | 120 | 0.83 | 0.20 | 0.10 | 1.19 | 11.90 |
| Hardwood sawdust | - | 66.5 | 0.3 | 221 | - | - | - | - | - |

Adapted from Jha et al. [37].

*2.3. Benefits of Biochar*

2.3.1. Interaction of Biochar in Soil

Biochar is known to sequester carbon and improve soil functions. Within a short period, the interaction between biochar, soils, microbes, and plant roots occurs after its incorporation into the soil [8]. The factors influencing the types of interactions are (i) feedstock composition, in particular, the total percentage and specific composition of the mineral fraction; (ii) pyrolysis process conditions; (iii) biochar particle size and delivery system; and (iv) soil properties and local environmental conditions. The aging of biochar starts before addition to soils and once incorporated, the rate is partly governed by the soil moisture and temperature condition [38]. Immediately after the application of biochar amendment, the evolution of biochar-derived carbon can be observed within the first 2 weeks and decreases exponentially with time [39]. Water plays a major role in mineral weathering processes such as hydrolysis, dissolution, carbonation and decarbonation, hydration, and redox reactions. The rate of these reactions depends on the type of biochar, nature of reactions, and pedoclimatic conditions. The dissolution and leaching of soluble salts (e.g., K and Na carbonates and oxides) present in the biochar is the first reaction among all the interactions. The dissolution makes the pH increase in the water film around the biochar particles [40]. The biochar converted from biomass is still thermodynamically unstable under the oxidative state of most surface soils [41]. Low-temperature biochar has a considerable fraction of non-aromatic C, which makes the biochar more susceptible to microbial attack [42] and subsequent oxidation than high-temperature biochar [43]. Despite the high stability of aromatic C, it has redox activity and functions as a reducing agent, $O_2$ being the most common electron-acceptor species. The electron-donating properties of an area with a high density of $\pi$-electrons boosted the abiotic reaction and initiated the oxidation of biochar [44]. The number of free radicles in biochar is dependent on the pyrolysis process [45] and thereby increases the reactivity towards the oxidation [46]. Biochar particles can have both acidic and basic properties and are greatly influenced by the moisture condition and the surface retention of ions through electrostatic interactions [47]. They usually co-exist, with the oxidative processes the concentration of basic sites decreases as the biochar particle weathers [48,49]. The biochar which has higher mineral content have interaction with organic matter and clay mineral surfaces depending on the type of clay (2:1, 1:1), distribution of functional groups on the clays (siloxane, OH), and organic matter (COOH, C=O, C–O, CN), the polarity of these compounds and the composition and concentration of cations and anions in solution [50]. There are also complex interactions between biochar with plant roots and microorganisms. Biochar interacts with the soils along with the root hairs. Once the root system encounters the biochar particle, the root hairs can penetrate the water-filled macropores of the particle and the organic compounds (including low- and high-molecular-weight compounds such as free exudates and mucilage); sloughed-out cells and tissues; and lysates from the growing root can be absorbed by biochar surfaces [51]. The fauna (such as worms, termites, larvae, and other insects) present in soil ingest or live inside biochar by breaking it up or coating it with organic compounds. Bioturbation by earthworms plays an important role in the physical mixing of the soil profile with the biochar. Over time, the downward movement of biochar increases within the soil profile where the soil microbial activity is lower [52].

2.3.2. Role of Biochar in Carbon Sequestration

An increase in ambient temperature has now been unequivocally proven and reported to increase at an unprecedented rate [2]. Since the late nineteenth century, global surface temperatures have increased by 0.88 °C [53]. Carbon dioxide ($CO_2$), methane ($CH_4$), and nitrous oxides ($NO_2$) are considered to be the important anthropogenic GHGs, which are released into the atmosphere through the burning of fossil and biomass fuels as well decomposition of above- and belowground organic matter. As per the report, carbon dioxide ($CO_2$) concentration has increased from 280 ppmv in 1850 to 380 ppmv in 2005 (up to 31% increase) [53]. An increase in the concentrations of methane ($CH_4$) and nitrous

oxide ($N_2O$) have also been observed over the same period but at a steady rate [2,53,54]. According to Pacala and Socolow [55], approximately 7 $PgCyr^{-1}$ (1 Pg = $10^{15}$ g) is emitted by fossil fuel combustion and around 1.6 $PgCyr^{-1}$ through deforestation, land-use change, and soil cultivation which in turn plays an important role in contributing to climate change leading to global warming. Thus, there lies a strong quest for mitigating the risks of global warming by stabilizing the GHGs present in the atmosphere [56–58]. As per Lal [9], three strategies can be adopted to lower $CO_2$ emissions *viz.* (i) reducing global energy use, (ii) developing low- or no-carbon fuel, and (iii) sequestering $CO_2$ from point sources or the atmosphere through natural and engineering techniques [58]. From the view of $CO_2$ sequestration, there is a wide range of processes and technological options available in agricultural, industrial, and natural ecosystems which include biotic and abiotic sequestration [9]. Studies have considered the potential of bio-based carbon materials for gas capture and storage, and biochar has emerged as one of the important tools among different carbon sequestration techniques [26,59,60]. The application of biochar in the soil poses a novel approach to establishing a significant long-term sink of atmospheric carbon dioxide ($CO_2$) in terrestrial ecosystems. With the use of a wide variety of biochar application programs, an estimation of 9.5 BTof carbon can be potentially stored in the soils by the year 2100 [26]. About 50% of the carbon can be sequestered during the conversion of biomass carbon to biochar as compared to only 3% carbon retention in soil by burning and less than 10–20% (after 5–10 years) through biological decomposition, thereby giving higher yields of stable soil carbon in soil upon application [26]. The recalcitrance mechanism in biochar is considered to be one of the most important phenomena for sequestering carbon for a longer period [35]. Long-term carbon sinks of biochar are also due to slow microbial decomposition and chemical transformation [16]. Biochar amended at 2, 5, 10, 20, 40, and 60% *w/w* levels corresponding to field application rate of 24–720 Mg $ha^{-1}$ has been reported to reduce $CO_2$ production as well as significant suppression of the ambient $CH_4$ oxidation and $N_2O$ production at all levels as compared to unamended soils [59]. Thus, biochar can offer both large and long-term C sink in the soil making it one of the desirable choices for carbon sequestration for mitigating climate change. The figure (Figure 1) below represents the mechanism through which biochar acts as a carbon sink. Thus, biochar production from biowaste can not only act as a promising precursor for $CO_2$ sequestration but also has also emerged as a sustainable strategy for solid waste management [60].

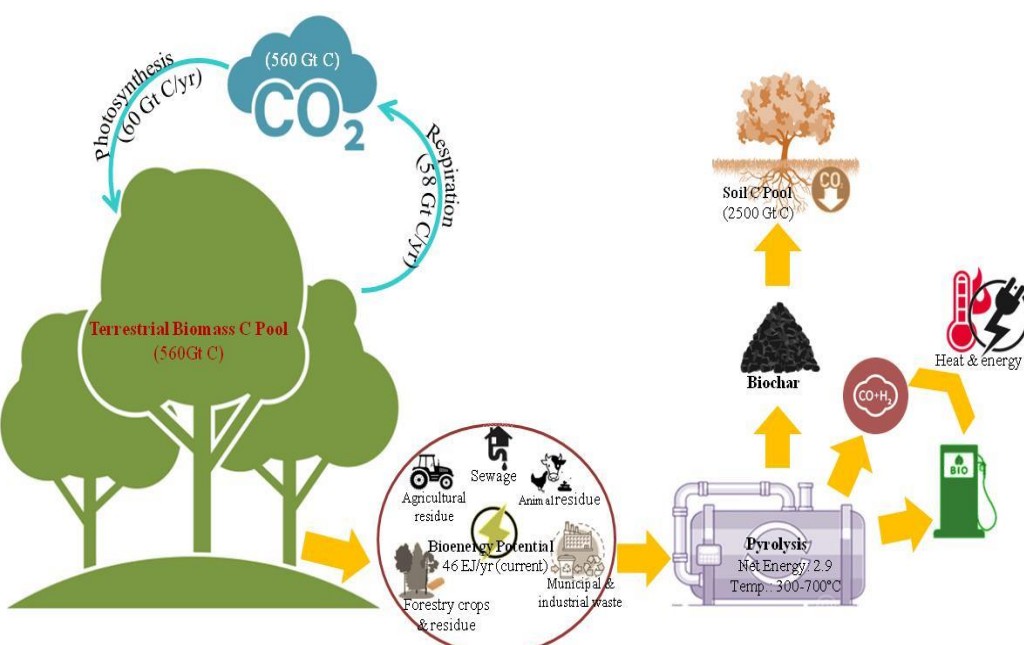

**Figure 1.** Schematic diagram of biochar-induced carbon sequestration.

### 2.3.3. Impact of Biochar on Soil Physical, Chemical, and Microbial Properties

Improved soil physical, chemical, and biological properties are desirable for optimum plant growth and development. Applications of biochar in soil are known to have a significant impact on various properties of soil [17,61,62]. The high porosity of biochar tends to improve a wide range of soil physical properties such as total porosity, soil density, soil moisture content, water holding capacity, and hydraulic conductivity [17,63,64]. Improvement in water retention capacity of the soil is mainly attributed to improved soil texture and aggregation posed by higher surface area and porosity of biochar [25,65,66]. Biochar applications at higher rates significantly increase the field capacity of the soil [67]. The effects can be more pronounced in non-irrigated regions with an increase in available water for crop growth as well as reducing the occurrence of water stress in between the events of rainfall. Addition of biochar decreases soil bulk density [62,68] which affects the infiltration rate in soil. Improved bulk density due to increased soil porosity will have a positive impact on soil aeration which is desirable for root and microbial respiration. Higher organic carbon content [69], as well as surface charge [70] in biochar, is another aspect that plays a crucial role in enhancing soil aggregation and its stability [69]. The stable soil aggregates change the structure of soil and thus improve soil moisture retaining capacity, infiltration, run-off reduction, and erosion [69].

Biochar plays a significant role in improving soil's chemical properties which includes raising pH, organic carbon, and exchangeable cations [27]. Most studies reported an increase in soil pH upon biochar additions [62,65]. An increase in cation exchange capacity (CEC) is confirmed by Lehmann et al. [71] which is an important property to prevent leaching loss of nutrients and thus can increase the fertilizer use efficiency (FUE). The higher CEC of biochar is reported to possibly enhance soil aggregation by aiding in forming certain complexes between organic matter and other minerals with that biochar [70]. However, it is observed that the effective cation exchange capacity is reported to increase with time after being incorporated into the soil [70]. This is so because the surfaces of biochar tend to get oxidized after getting in contact with moisture (water) and air [48,70,72]. The advantages of an increase in pH value on biochar application are more pronounced, especially in acidic soils that are associated with heavy metal toxicity or nutrient deficiencies. Depending on the pH-buffering capacity of the soil, biochar is reported with typical high liming equivalence in raising the pH value in acidic soils [18]. The increase in pH due to liming effect of biochar can play a significant role in the availability of essential nutrients in the soil. Important macro (N, P, K, Ca, Mg) and micro-nutrients (Cu, Fe, Mn, Zn,) which are essential for plant growth and development are reported to increase upon application of biochar in soil [27,29,73,74]. Apart from this, biochar due to its high affinity to hold nutrients reduces nutrient loss through leaching which in turn increases fertilizer use efficiency by the plant [71]. Several investigations have confirmed that volatilization of $NH_4^-$ decreases significantly with a high biochar application rate (10% or 20%, $w/w$) due to high CEC [71] however, biochar with high N content may lead to a higher leaching of $NO_3^-$ [31]. Biochar particles are assumed to act like clay and thus hold large amounts of immobile water even at increased matric potentials. Several other studies also reveal that the addition of biochar significantly increases the nodulations of rhizobia [75] thereby confirming the improvement in nitrogen fixation [76]. As per Biederman and Harpole [75], the increase in N-fixation following biochar application was reported to be 72%. In terrestrial ecosystems, biochar is also observed to act as a habitat for mycorrhizal fungi. The porous structure of biochar provides a habitat for microbes in soil and protects them from predation [61]. The habitat leads to a ubiquitous symbiotic association between them and favors the soils to carry out various ecosystem services in contributing to sustainable plant production and ecosystem restoration [61].

However, changes in soil properties as discussed above depend on biochar application rate, type of feedstock, soil type, pyrolysis parameters, and various other conditions prevailed [27,77,78]. Several studies revealed that biochar, when applied at sufficiently high rates tends to improve soil's physical properties [18,27,79,80]. Chan et al. [28] reported

that the increasing rate of biochar application tends to increase the field capacity but, the significant changes could only be observed at higher rates of biochar application, i.e., 50 and 100 Mg ha$^{-1}$. Soil amended with biochar made from green waste with application rates of 50 and 100 Mg ha$^{-1}$ to an alfisol has shown significant retention of water at field capacity compared to control [27]. In another case, the addition of mixed hardwood biochar at 1 and 2% (*w/w*) to a mollisol, did not detect any effect on moisture retention at soil water potential of $-0.33$ bars and $-15$ bar however, significant increases in moisture retention were observed at $-1$ and $-5$ bars soil water potential compared to control [62]. An increase in pH value after application of biochar is reported to be higher in sandy and loamy soils as compared to clayey soils [81] however; buffering capacity is reported to be higher in finely textured soil compared to that of coarse-textured soil. Nutrient retention capacity is found to be higher in aged biochar when compared to fresh biochar [48], which suggests that CEC increases in soil over time following biochar application. It can act as an alternative to fertilizer. Thus, biochar can be effectively used as a soil amendment to improve its overall quality in a sustainable, economic, and environmentally friendly way. The overall effect of biochar on various soil properties has been represented in Figure 2.

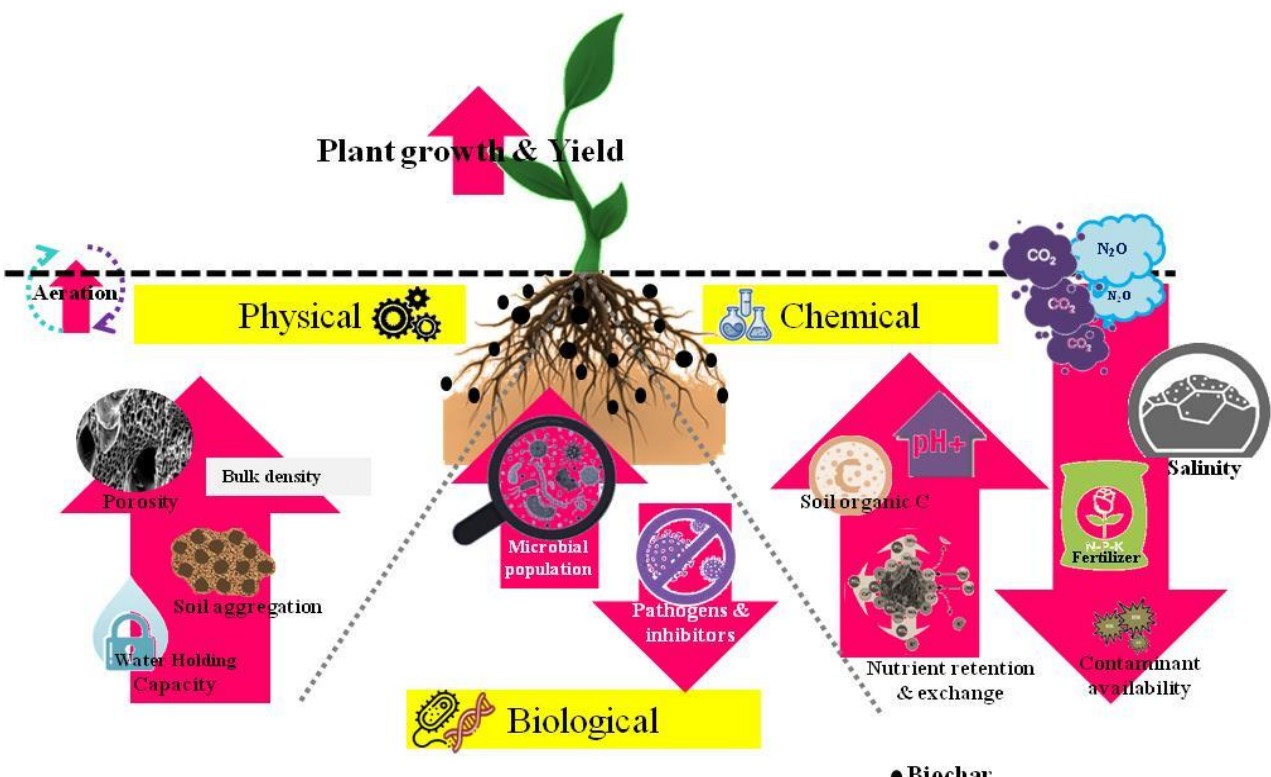

**Figure 2.** Graphical representation of the overall effect of Biochar after soil application.

## 3. Biochar as Soil Amendments

Biochar can act as a soil amendment as it can improve soil quality through altering various physiochemical and biological aspects of soil. Apart from this, its ability to absorb various contaminants (organic and inorganic) from the soil due to the large surface area and aromatic-carbon content biochar can serve as an important and desirable property for bioremediation of the polluted soil [82]. Biochar provides favorable habitat for soil microbes owing to its influence on soil organic carbon content (SOC) as well as water retention capacity [60]. Changes in the nutrient and C availability in the soil can also provide physical protection to beneficial microorganisms [83]. These properties have led to the renewed interest of scientists in using biochar as one of the important soil amendment techniques in the stabilization of soil organic matter. Thus, biochar as soil amendments can potentially increase conventional agricultural productivity and enhance the farmer's

ability to participate in carbon markets beyond the traditional approach where the carbon is directly applied to the soil [84]. Several findings have reported the positive response of crops following biochar applications apart from reducing GHGs [34,85]. According to Liu et al. [84], the application of biochar at the rate of 40 and 30 Mg ha$^{-1}$ has significantly increased the yields of rapeseed and sweet potato along with improvement in soil structure, macroaggregates content, and stability of aggregates. Simultaneously, biochar amendments to soil have been reported to increase SOC, total N, and C:N ratio. As per the experiment conducted by Major et al. [34] for 3 years, it was reported that biochar applied at 20 Mg ha$^{-1}$ showed 28, 30, and 140% increase in the yield of maize grain over control for the years 2004, 2005, and 2006, respectively. The report further showed an increase in soil nutrients such as Ca and Mg and pH value. Slow release of nutrients, stabilization of organic matter, and increased retention of nutrients in soil over time are observed to be the long-term effects of biochar application [32,86]. Application of biochar along with fertilizers in agricultural soils can produce significant benefits for the growth and yield of crops [24]. Higher grain yields of upland rice grown in P deficit soil conditions were reported by Asai et al. [25] indicating the positive response of biochar application in combination with chemical fertilizer. Application of biochar in the soil can also increase drought tolerance and water use efficiency [87]. Thus, the properties of biochar can play a vital role in the promotion of sustainable agriculture [74]. However, to identify the most suitable application of biochar as a soil amendment, it is very crucial to understand the physicochemical properties of biochar [77,78]. The occurrence of specific unfavorable crop and soil combinations is reported to have negative effects on crops [63].

*Nutrient Content and Its Transformation*

The nutrient content and transformation of biochar is an important factor to consider. The N content in biochar is very low due to its sensitivity to temperature while S is slightly depleted for high thermal treatment [87]. During pyrolysis, some nutrients are caused to volatilize especially at the surface of the material, while other nutrients become concentrated in the biochar matrix. The chemical properties of biochar are directly influenced by temperature. During the heating process, individual elements are lost to the atmosphere, fixed as recalcitrant forms, or liberated as soluble oxides. In the case of wood-based biochar formed under natural conditions, C begins to volatilize around 100 °C, N above 200 °C, S above 375 °C, and K and P between 700 and 800 °C. The volatilization of Mg, Ca, and Mn occurs at temperatures greater than 1000 °C [88,89]. Biochar produced from sewage sludge pyrolyzed at 450 °C contains over 50% of the original N (although not in bio-available form) and all of the original P [90]. Pyrolysis of wheat straw at 500 °C results in a loss of about 50% of S [91,92], and the concentration of $NH^{4+}$ and $PO_4^{3-}$ decreases with the increase in pyrolysis temperature [93]. Biochar acts as a soil conditioner and driver of nutrient transformations and as a primary source of nutrients [30,71]. The availability of nutrients to plants is directly dependent on nutrient transformation which thereby influences plants' productivity [94]. Biochar plays a significant role in the transformation of nutrients through microbes. Deluca et al. [95] observed that biochar incorporation into soil increased the rate of nitrification in forest soil. It also increases the abundance of ammonia-oxidizing bacteria [96]. Biochar application may alter or even reduce net N mineralization [95] which might lower the available N for plants [71]. The activity of alkaline phosphatase, aminopeptidase, and N-acetylglucosaminidase was increased with the application of biochar [97,98] while there is a decrease in cellobiosidase and glucosidase. The growth of fine roots and root hairs into biochar pores stimulates the production of organic N and P mineralizing enzymes. Lehmann et al. [83] observed that biochar induces changes in the bacterial community suggesting a broader effect due to plant nutrient uptake as an explanation for the greater enzyme activity.

## 4. Method of Production, Storage, and Incorporation of Biochar

Biochar can be made from a wide range of organic sources at varying temperatures in the limited or absence of air. Yao et al. [40] produced dry-pyrolysis biochar at three different temperatures (300, 450, and 600 °C) through slow pyrolysis inside a furnace under an $N_2$ environment. The samples were then sieved to a uniform size fraction of 0.5–1 mm and washed with deionized (DI) water several times to remove impurities. The samples were oven-dried (80 °C) for about 12 h and stored for later use. If the produced biochar comes in contact with the moisture during its storage then the initial weathering of biochar particles may occur [46]. This phenomenon is known as 'aging' and occurs as a result of the oxidation of exposed C rings with a high density of π-electrons [44] and free radicals [99]. Addition of decomposable organic material such as compost to biochar might accelerate weathering reactions [100,101]. The method of biochar incorporation into the soil may modify the structure and particle size of biochar which thereby will affect water holding capacity and mineralization rate [102]. Ploughing results in greater soil mechanical disturbances than any other methods such as deep banding or direct drilling which stimulates the decomposition of biochar [103]. Spreading and incorporation with rotary plough, and deep banding are the most common methods that involve the incorporation of large volumes of biochar (>5 t) into the soil to a depth of 60–100 mm [2,104].

## 5. Finding Possibilities of Biochar Introduction to Hill Ecosystem

### 5.1. Potential Role in Sustainable Agriculture

In the EHR of India, the main form of agriculture is Shifting or "slash and burn" cultivation. Locally the cultivation practice is known as "jhum" and is strongly based on traditional knowledge. It was considered an appropriate sustainable land use practice in diverse socio-economic setups where the dependent human population was within the carrying capacity of a 10–15 year jhum cycle. However, due to the rising population and increasing food demand, the jhum cycle in most areas has been reduced to 2–3 years and today, scientists view shifting cultivation as environmentally destructive and a faulty land use practice having a very low output–input ratio. One of the most important negative impacts of jhum cultivation in the region is the acceleration of soil erosion besides causing environmental pollution due to burning. It is reported that around 38–84% of the biomass C of vegetation is released due to burning in most of the shifting cultivation systems [105]. Even then, jhumming is unavoidable in the region due to cultural ethos and has been the way of life since time immemorial. Under this condition, where shifting cultivation is dominantly practiced, biochar is identified with huge opportunities [26]. Application biochar will commonly serve as a component of soil organic matter, especially in the region where slash-and-burn agricultural practices are dominant [106]. Biochar can play a vital role in carbon sequestration by converting the traditional practice of "slash and burn" into "slash and char" (as shown in Figure 3). Around 0.21 Pg of carbon (12% of the total anthropogenic carbon emissions by land-use change) can be sequestered in the soil if the traditional "slash and burnt" practice is converted to "slash and char" [22,107]. The slash and char method can also potentially offset the total anthropogenic emissions of carbon by up to 12% annually [37]. Producing biochar through simple kiln techniques and its application in the soil can sequester more than 50% of the C in a highly stable form [108]. Thus, biochar can play a vital role in carbon sequestration for mitigating climate change, especially in hills and mountain agriculture.

The EHR is bestowed with a tremendous amount of diverse vegetation. The region is rich in weed diversity which is most prevalent in the agricultural field as well as in the forest. Part of agricultural residue, forest wood, litter, kitchen, industrial waste, etc., is left unused (Table 2) while a major proportion is burnt directly. Although direct burning provides easy waste management practice, it tends to create severe air pollution in the surrounding region. In addition, the topography of most of the EHR is dominantly characterized by a steep slope and deep terrain which serve as one of the biggest factors in limiting most of the agricultural and allied activities. Thus, most of the area remains unused and gets

included under wasteland. The unpalatable scenario creates a tremendous amount of opportunity for biochar to come into play in tackling the issues that prevailed in the region. The abundance of vegetation in the region provides a large opportunity to develop a wide variety of biochar (Table 2). The development of biochar can play a vital role in recycling a wide variety of unused waste material which can be substantially incorporated as soil amendments for improving the soil fertility status in addition to carbon sequestration.

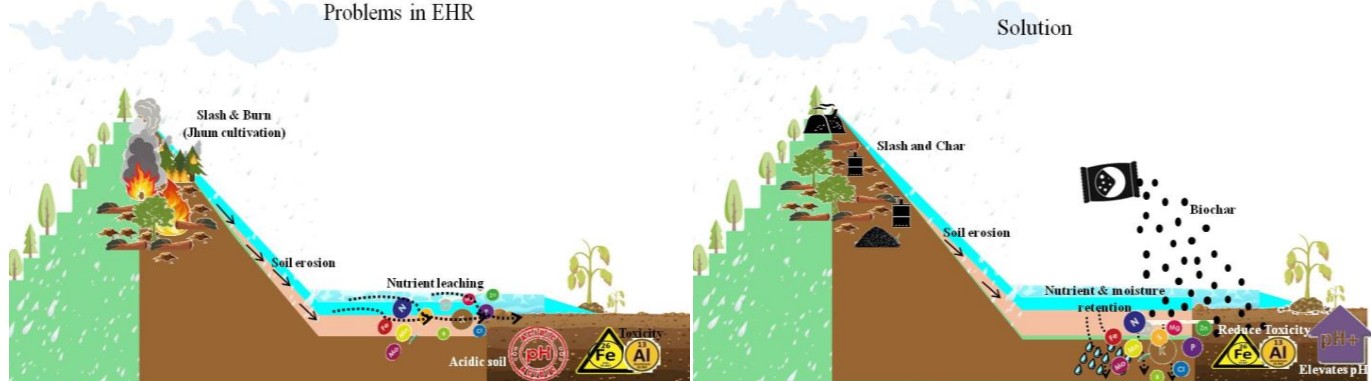

**Figure 3.** Agriculture issues and the potential solution in EHR.

**Table 2.** Generation and surplus of crop residues in different states of EHR of India (000′ Mg ha$^{-1}$).

| States | Residue Generation [109] | Residue Surplus [109] | Residue Burned [110] | Residue Burned [111] |
|---|---|---|---|---|
| Arunachal Pradesh | 400 | 70 | 60 | 40 |
| Manipur | 900 | 110 | 140 | 70 |
| Meghalaya | 510 | 90 | 100 | 50 |
| Mizoram | 60 | 10 | 10 | 10 |
| Nagaland | 490 | 90 | 110 | 80 |
| Sikkim | 150 | 20 | 10 | 10 |
| Tripura | 40 | 20 | 220 | 110 |
| Total | 2550 | 410 | 650 | 370 |

As far as the soil of EHR is concerned, around 95% are acidic, and nearly 65% are suffering from strong acidity (pH < 5.5) [6]. This is one of the serious issues that hamper crop production in the region. The lack of irrigation facilities is another factor contributing to lower yield as the nature of farming in most of the hilly region are rainfed. In addition, high rainfall in the region is also a factor that has plagued crop production. As heavy in the region is responsible for leaching of nutrients which leads to deficiency of P, Al toxicity and various environmental issues [112]. Since biochar is known to act as a limiting material that significantly raises the pH value [65] as well as improves the soil water-holding capacity, there is large scope for biochar in the region as it can play a vital role in ameliorating the soil acidity as well as reducing the requirement of irrigation inputs. It can also efficiently adsorb $NH_3$, $NO_3$, $PO_4^{3-}$, other ionic solutes [77,108,113,114] as well as hydrophobic organic pollutants [115] thereby reducing environmental pollution. The properties of biochar to act as a buffer for ammonia in the soil gives the potential to decrease ammonia volatilization from agricultural fields. Thus, biochar can largely enhance the nutrient use efficiency in the EHR.

The nature of agricultural practice in the EHRis organic by default, due to the absence or meager use of synthetic agrochemicals, diverse vegetation, suitable climatic conditions, use of indigenous knowledge, etc., Sustaining the long-term fertility status of soil, maintaining a healthy environment, conserving the diversity of the ecosystem, recycling the farm residue, etc., are some of the important components of organic agriculture. The long-term effect on soil fertility can be achieved following biochar application due to its recalcitrant property that is responsible for the stability of biochar [8]. Hence, biochar can

play a substantial role in promoting sustainable agriculture in the EHR. A few important supporting facts regarding biochar's suitability for the region have been given below:

- The contribution of slash and burn agriculture in increasing the air $CO_2$ concentration of the EHR cannot be ignored. Thus, the adoption of "Slash and Char: instead of "Slash and burn" can contribute immensely to reducing the GHGs in the EHR of India.

- In the hills of North East India, wood is burnt every day for cooking. Biochar could be produced commercially in 'roadside hotels' where a decent amount of wood is burnt every day by conventional methods. Energy-efficient smokeless stoves have been developed to produce biochar as a byproduct. Such a program is already underway in India and countries in Africa, South America, and the Asia Pacific. Programs on smoke-free and energy-efficient stoves have already been initiated in India and a general consideration on adding up the objective of producing biochar by such clean stoves program will fit better in our system.

- Soil erosion and nutrient loss are serious threats in the hill slopes of the North East. As biochar has higher surface area and is porous in nature, the application of biochar in soils of hill slopes can minimize soil erosion and nutrient loss.

### 5.2. Potential Feedstocks Reserve in the Region

Although several reports on biochar are available worldwide, there are fewer reports on biochar in the NE region. This creates an opportunity to develop research thrust on biochar among the scientific communities in the region. The development of low-cost, small-scale biochar production units can help to build a garden, agricultural, and forest productivity and also will help in providing thermal energy for cooking and drying grain in the region. Because of high rainfall in the regions, there is abundant availability of weeds (~100–120 Mg ha$^{-1}$). The major weed species available in this region are *Lantana Camara*, *Chromolaena odorata*, *Ageratum* sp., etc. Biochar was prepared from locally available weed biomass and was characterized by different authors have been presented in Table 3. Availability of 5–20 Mg ha$^{-1}$ of weed biomass in both cropped and non-cropped areas of the HER has been reported [116]. Rearing animals (poultry, cattle, pig, etc.) is a common agricultural practice adopted by the farmers of the states. Conservative estimates show that about 15 Mt/annum of animal dung and 9 Tg annum$^{-1}$ of crop residues are available in northeastern states [117]. After harvesting crops, the residues of rice, maize, mustard, pulses, etc., remained unutilized and can be effectively used for biochar preparation. The generation and surplus of crop residues in different states of North East India are given in Table 4. Apart from these biomass resources, forest biomass (forest cover >65%) provides alternative sources of raw materials for the production of biochar. Pine needles and barks available in the region (particularly in Meghalaya) can be a feedstock for biochar production. Moreover, there is ample opportunity for conversion of "slash and burn" to "slash and char".

**Table 3.** Biochar's properties derived from different weeds (at 400 °C) from EHR.

| Source/Biomasses | Productivity | MC | Total OM | VM | Total Organic | Ash | Ref. |
|---|---|---|---|---|---|---|---|
| *Brachariamutica* | 33.33 | - | - | 30.60 | - | 32.90 | Awasthi et al. [118] |
| Water hyacinth and Para grass | 58.2 | - | - | 39.44 | - | 30.50 | |
| *Saccharum ravannae* | - | 5.88 | - | 35.46 | - | 7.15 | Saikia et al. [119] |
| *Arundo donax* | 43.87 | 5.63 | 34.02 | 34.02 | - | 8.27 | Saikia et al. [120] |
| *Ageratum conyzoides* | 38.7 | 10.1 | 78.8 | 26.2 | 52.6 | 21.2 | |
| *Lantana camera* | 41.6 | 10.2 | 77.9 | 25.9 | 52.0 | 22.1 | |
| *Gynura* sp. | 35.9 | 10.3 | 79.9 | 25.9 | 53.9 | 20.1 | Mandal et al. [121] |
| *Setaria* sp. | 44.9 | 10.1 | 80.9 | 25.8 | 55.2 | 19.0 | |
| *Avenafatua* | 45.5 | 8.1 | 83.8 | 27.5 | 56.2 | 16.3 | |
| Pine needles | 48.2 | 9.0 | 76.9 | 22.3 | 54.6 | 23.1 | |
| *Ipomoea carnea* | 34.18 | - | - | 56.34 | - | 5.49 | Konwer et al. [122] |

**Table 4.** Estimate of crop residue production (% of gross) in EHR (2014).

| States | Cereals | Oilseeds | Sugarcane | Horticulture | Pulse | Others | State Average |
|---|---|---|---|---|---|---|---|
| Arunachal Pradesh | 27 | 11 | 33 | 25 | 22 | 10 | 21 |
| Manipur | 28 | 21 | 40 | 25 | NA | NA | 29 |
| Meghalaya | 26 | 15 | 40 | 20 | 35 | 30 | 28 |
| Mizoram | 29 | 18 | 40 | 32 | 35 | 48 | 34 |
| Nagaland | 27 | 16 | 40 | 43 | 35 | 38 | 33 |
| Sikkim | 28 | 22 | NA | NA | NA | NA | 25 |
| Tripura | 34 | 21 | 40 | 45 | 35 | 38 | 37 |
| | 28 | 20.5 | 39 | 33.2 | 34.8 | 35.3 | 248 |

Adapted from Hiloidhari et al. [115].

Biochar was prepared from locally available weed biomass at 400 °C and showed the presence of more than 50% stale carbon. Mandal et al. [121] applied biochar produced from the weeds in maize alone or combination with fertilizers. From the experiment conducted it was observed that biochar application improved soil pH by 0.26–0.30 units within two months of its application. All the biochar has a significant impact on SOC and ranged between 1.7 and 1.74% as compared to 1.62% in control. The highest increase in SOC was observed with biochar of *Gynura* sp. followed by biochar of *Ageratum, Lantana*, and *Seteria*. Alone application of biochar elevated available nitrogen (4.5 to 21.3 mg $kg^{-1}$), available P (3.32 to 3.68 mg $kg^{-1}$), and K content by 20% above control. The result obtained emphasized the potential of weed biomass as an alternative to enhance soil and crop productivity and its conversion from waste to resource. SOC, SMBC, and activity of dehydrogenase enzyme of surface soil (0–15 cm) were improved markedly with the application of biochar. The study also revealed that exchangeable aluminum was reduced substantially due to the application of biochar. Thus, biochar may offer a win-win technology for a sustainable crop production system.

## 6. Biochar Initiatives in India

RaGa LLC in collaboration with two NGOs in India, Appropriate Rural Technology Institute (ARTI) and Janadhar, is a pioneering decentralized biochar production organization that uses modular pyrolysis kilns in villages and small towns in India for the production of biochar from sustainable resources such as waste biomass, organic municipal solid waste (MSW) and bagasse (during harvest season).Latur, a medium-sized city in Western India, and nearby towns will be the site of their first biochar operation. In the 1st phase, ARTI technology—locally manufactured kiln-and-retort systems that are easily assembled and deployed quickly—will convert the organic waste into biochar. Jalandhar will manage the day-to-day operations of the production facilities. These groups also will conduct training for farmers to help them test biochar as a soil amendment and perform pot and field trials. For the 2nd phase, RaGa is evaluating other pyrolysis technologies for the mass production of biochar.

The Society of Biochar India was formally registered as an NGO in 2010. The NGO works to support and highlight the project associated with biochar in India. Various small-scale biochar kilns have been developed to facilitate the production of biochar at a local level in the vicinity of a village which will cut the cost of transportation of feedstock and the setting and maintenance cost of a large-scale kiln. In this regard, Holy Mother Biochar Kiln was developed by Sarada Matt (Holy Mother) at Almora, Uttarakhand; "Drum Kiln" was developed by Venkatesh et al. [123] by modifying oil drums at ICAR-Central Research Institute for Dryland Agriculture (CRIDA), Hyderabad; ICAR- Central Institute of Agricultural Engineering (CIAE). Portable Charring Kiln developed by the researcher at ICAR-CIAE, Bhopal; and portable metallic kiln by ICAR Research Complex for North Eastern Hill Region, Umiam is economically and environmentally viable option [124].

## 7. Potential Negative Impacts of Biochar

There is little knowledge about the toxic effects of biochar incorporation into the soil. Mukherjee and Lal [18] reviewed the negative aspects of biochar amendment on crop yield, soil quality, and associated financial risk as many heavy metals and plant-available organic compounds may be present on their surfaces during its synthesis. These toxic compounds are condensed as polycyclic aromatic hydrocarbons (PAHs), cresols, xylenols, formaldehyde, acrolein, and other toxic carbonyl compounds that have bactericidal or fungicidal activity [125]. The residual volatile on biochar becomes a flash carboniser which is toxic to plants [126]. However, these can be removed with the soaking of biochar with sufficient time. Biochar produced from sewage sludge and tannery residue generally contained heavy metals at high levels (e.g., Cu, Cr, Zn, etc.).

Keiluweit et al. [127] quantified eleven unsubstituted three- to five-ring PAHs and alkylated forms of anthracene and phenanthrene in wood and grass biochars produced in a temperature range of 100 to 700 °C. The high sorption capacity of biochar may limit the leaching of PAHs from soils. In addition, the presence of persistent free radicals (PFRs) on the surface can inhibit the germination and growth of rice, wheat, and corn seedlings [128]. Alkaline biochars may limit the availability of specific soil nutrients and have a negative influence on soil cation exchange capacity [129]. As per the experiment conducted by Lee et al. [130] found that pH would influence cation exchange capacity, ranging from $-10$ to 30 cmol kg$^{-1}$ with the pH increasing from 5.0 to 8.5. Additionally, biochar with high pH is always accompanied by electrical conductivity which results in the unavailability of nutrients to plants. Herath et al. [131] observed that application of biochar at higher rates leads to clogging of micropores by mineral/ash which adversely affects the water holding capacity of the soil.

## 8. Conclusions

Hills and the mountainous region of India pose a vast opportunity for the production and application of biochar. It can potentially improve soil properties and can serve for the growth and development of the farming communities in the hilly region in addition to mitigating climate change through carbon sequestration. With the advancement in knowledge about biochar in agricultural productivity, the effects of biochar on biological conditions such as microbial, faunal, and root abundance, community composition of various biota and their functions, chemical processes such as nutrient content, weathering of soils, etc., and physical processes such as structure, texture, etc., were justified. Today's important crisis is the increasing level of GHGs ($CO_2$, $CH_4$, and $NO_2$) in the atmosphere which disturbs the environmental system in such a way that all living creatures could become extinct one day due to global warming. The addition of biochar from any organic sources can sequester C in the soil and decrease the rate of the release of GHGs. Biochar also enhances the soil conditions so that agricultural productivity increases and the ecosystem is restored.

However, the suitability of biochar type and its response to specific crops needs to be identified before application. The less reporting on changes in soil properties upon biochar application in the region also provides research priorities among the scientific communities. Further research needs to be carried out to understand the full mechanisms involved in changes in soil physicochemical, hydrological, and ecological properties in the long term.

**Author Contributions:** Conceptualization, J.L. and R.N.; methodology, J.L. and R.N.; software, J.L. and K.R.; validation, J.L., R.G.I. and S.B.; formal analysis, J.L. and A.D.; investigation, J.L., K.R. and S.M.; writing—original draft preparation, J.L., R.N. and S.D.; writing—review and editing, J.L., S.S., S.B., A.B., K.R. and A.D.; visualization, S.H.; supervision, B.U.C., S.H. and V.K.M.; project administration, S.H. All authors have read and agreed to the published version of the manuscript. Please turn to the taxonomy for the term explanation.

**Funding:** This research received no external funding.

**Institutional Review Board Statement:** Not applicable.

**Informed Consent Statement:** Not applicable.

**Data Availability Statement:** Not applicable.

**Acknowledgments:** Authors sincerely acknowledge the National Innovations in Climate Resilient Agriculture (NICRA) project (PIMS Code# OXX01713) of the Indian Council of Agricultural Research (ICAR) for facilitating the biochar work.

**Conflicts of Interest:** The authors declare no conflict of interest.

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
