# Peer review of "Prospects of Biochar for Sustainable Agriculture and Carbon Sequestration: An Overview for Eastern Himalayas"

_sustainability, doi:10.3390/su14116684_

Round 1

Reviewer 1 Report

The present article " Prospects of biochar for sustainable agriculture and carbon sequestration: an overview for Eastern Himalayas" efficiently provided the detailed information about role of biochar for altering the soil properties for sustaining agriculture and carbon sequestration especially for hills and mountain ecosystem. The article is well written and effectively represented. However, in order to further consider for publication, the article needs some minor revision. The following comments must be addressed by the authors:

*   Introduction needs refining. The author has not cited some previous published articles on charcoal/biochar in introduction section explaining its role in Syngas production (gasification). Some mentioned references should be cited: https://doi.org/10.1016/j.jcou.2022.101890; https://doi.org/10.1002/ceat.201900547.

* Authors should explain the difference between this manuscript with previous publication entitle: ” Prospects of Biochar for Sustainable Agriculture and Carbon Sequestration in North Eastern Hill Region of India”

* Text of the figures should be clear. Please redraw these.

* The English language needs more refining. The manuscript should be proofread to reduce the chances of grammatical errors. The author is requested to go through the whole of the manuscript and rectify them.

Author Response

Comments and Suggestions for Authors

The present article "Prospects of biochar for sustainable agriculture and carbon sequestration: an overview for Eastern Himalayas" efficiently provided the detailed information about role of biochar for altering the soil properties for sustaining agriculture and carbon sequestration especially for hills and mountain ecosystem. The article is well written and effectively represented. However, in order to further consider for publication, the article needs some minor revision. The following comments must be addressed by the authors:

Answer: Thank you for reviewing the manuscript and giving us the appreciation about making this review paper.

  1. Introduction needs refining. The author has not cited some previous published articles on charcoal/biochar in introduction section explaining its role in Syngas production (gasification). Some mentioned references should be cited: https://doi.org/10.1016/j.jcou.2022.101890; https://doi.org/10.1002/ceat.201900547.

Answer: Thank you for the appreciation about making this review paper. As per suggestion, we have refined the introduction part of this manuscript.

As per referee suggestion, we have provided the relevant article charcoal/biochar in introduction section explaining its role in Syngas production (citation 10).

[10] Siegelman, R.L., Milner, P.J., Kim, E.J., Weston, S.C. and Long, J.R., 2019. Challenges and opportunities for adsorption-based CO 2 capture from natural gas combined cycle emissions. Energy & environmental science, 12(7), pp.2161-2173.

We have also read all the articles (https://doi.org/10.1016/j.jcou.2022.101890; https://doi.org/10.1002/ceat.201900547) and found very interesting and highly relevant to the current research work. The important information extracted from these articles is added into the revised manuscript.

References:

[61] Karimi, M., Shirzad, M., Silva, J.A. and Rodrigues, A.E., 2022. Biomass/Biochar carbon materials for CO2 capture and sequestration by cyclic adsorption processes: A review and prospects for future directions. Journal of CO2 Utilization57, p.101890.

[93] Uddin Monir, M., Abd Aziz, A., Vo, D.V.N. and Khatun, F., 2020. Enhanced hydrogen generation from empty fruit bunches by charcoal addition into a downdraft gasifier. Chemical Engineering & Technology, 43(4), pp.762-769.

The same has been included in the revised manuscript.

  1. Authors should explain the difference between this manuscript with previous publication entitle: “Prospects of Biochar for Sustainable Agriculture and Carbon Sequestration in North Eastern Hill Region of India”

Answer: We made a Technical Bulletin (hard copy) entitled “Prospects of Biochar for Sustainable Agriculture and Carbon Sequestration in North Eastern Hill Region of India” from the results of our Biochar project in ICAR Research Complex for North Eastern Hill Region, Umiam, Meghalaya. The content of our review paper is different from that of our Technical Bulletin although there is some similarity in the title of the both.

  1. Text of the figures should be clear. Please redraw these.

Answer: We are thankful to the reviewer for this important comment. As per referee suggestion we have replaced the figures with ones with proper visibility.

  1. The English language needs more refining. The manuscript should be proofread to reduce the chances of grammatical errors. The author is requested to go through the whole of the manuscript and rectify them.

Answer: As per referee suggestion, outmost care has been taken to improve the quality of English used in the manuscript and accordingly corrections has been incorporated as and when required.

Reviewer 2 Report

The authors presented an overview of the prospects of biochar and carbon sequestration (Eastern Himalayas).  The article is written in an acceptable manner and has valuable content for the Journal's audience. The article is simple but concise and thorough and should be considered a good read as well. The strength of the paper is in the merge of the scientific overview into the practical sphere.  The review is clear, and is relevant to the field. The references are current and satisfying. The statements are coherent and supported by listed citations. Figures are generally ok, but one presented (Figure 3) should be improved.

I suggest minor changes in the paper:

L16 - please change from "others form" to "other forms" 

L21 - please change from "... capacities and advances..." to "capacities,(COMMA) and advances..."

L28 - please change from "... time and place and transportation..." to "time and place and,(COMMA) transportation."

L28 - "issues.." - remove the second dot from the sentence

L80 - Joule is always meant to represent a unit for energy. I suggest replacing "terra joule of energy" with just "TJ". Your readers are familiar with the basics.

L208 - why is "Biochar" under quotation marks? I suggest removing them.

L414 - I suggest that you improve the quality of Figure 3., it is not suitable for the quality of the Journal.

L444 - "...value [65]as well as..." - insert spacing after the reference

L448 - "It can also efficiently adsorb NH3 nitrate, phosphate, other ionic solutes" - use either names or formulas, consistently.

L492 - "about 15 Mt/annum of animal dung and 9 Tg per annum" - decide if you are using "/"; "^-1" or "per" and use it consistently throughout the paper

L578 - "moutainors" - spelling problem

L578 - "Hills and moutainors region of India poses..." either a region "poses" or Hills and mountain region "pose"

L579 - "ofbiochar" - spelling problem

L579 - " It can potentially improve soil properties and can serve for the" - I suggest removing the second use of the word "use"

L585 - I suggest removing "crucially" from the sentence, it diminishes the point.

Author Response

Comments and Suggestions for Authors

The authors presented an overview of the prospects of biochar and carbon sequestration (Eastern Himalayas).  The article is written in an acceptable manner and has valuable content for the Journal's audience. The article is simple but concise and thorough and should be considered a good read as well. The strength of the paper is in the merge of the scientific overview into the practical sphere.  The review is clear, and is relevant to the field. The references are current and satisfying. The statements are coherent and supported by listed citations.

Ans. Thank you very much for thoroughly reviewing the manuscript for its improvement and appreciating our work.

  1. Figures are generally ok, but one presented (Figure 3) should be improved.

Answer: As per reviewer suggestion Figure 3 have been replace with a better-quality image.

I suggest minor changes in the paper:

  1. L16 - please change from "others form" to "other forms" 

Answer: As per reviewer’s suggestion, the error has been rectified in the revised manuscript (Line no# 16).

  1. L21 - please change from "... capacities and advances..." to "capacities, (COMMA) and advances..."

Answer: As per suggestion, we have inserted comma in the place suggested in the in the revised manuscript (Line no# 21).

  1. 4. L28 - please change from "... time and place and transportation..." to "time and place and, (COMMA) transportation."

Answer: As per reviewer’s suggestion, we have inserted comma in the place suggested in the in the revised manuscript (Line no# 28).

  1. L28 - "issues.." - remove the second dot from the sentence

Answer: As per suggestion the error has been rectified in the in the revised text (Line no# 28).

  1. L80 - Joule is always meant to represent a unit for energy. I suggest replacing "terra joule of energy" with just "TJ". Your readers are familiar with the basics.

Answer: We are thankful to the reviewer for this important comment. As suggested by the reviewer we have taken the necessary action and instead of “terra joule of energy”, we used "TJ" in the revised text (Line # 88-89).

  1. L208 - why is "Biochar" under quotation marks? I suggest removing them.

Answer: As per suggestion, we have removed the quotation mark for Biochar in the revised manuscript.

  1. L414 - I suggest that you improve the quality of Figure 3., it is not suitable for the quality of the Journal.

Answer: As per suggestion, the Figure 3 has been replaced with better quality figure.

  1. L444 - "...value [65]as well as..." - insert spacing after the reference

Answer: As per suggestion, we have inserted spacing after the reference in the revised manuscript (Line # 460).

  1. L448 - "It can also efficiently adsorb NH3 nitrate, phosphate, other ionic solutes" - use either names or formulas, consistently.

Answer: As per suggestion, we have incorporated the above suggestions in the revised manuscript (Line #463)

  1. L492 - "about 15 Mt/annum of animal dung and 9 Tg per annum" - decide if you are using "/"; "^-1" or "per" and use it consistently throughout the paper

Answer: As per referee suggestions we have taken care to maintain consistency regarding the mention of units in the revised manuscript (Line # 508 and other places).

  1. L578 - "moutainors" - spelling problem

Answer: The error has been rectified as per the suggestion (Line # 584).

  1. L578 - "Hills and moutainors region of India poses..." either a region "poses" or Hills and mountain region "pose".

Answer: The error has been rectified as per the suggestion (Line # 584 in the revised manuscript)

  1. L579 - "of biochar" - spelling problem

Answer: The error has been rectified as per the suggestion (Line # 585 in the revised manuscript)

  1. 15. L579 - " It can potentially improve soil properties and can serve for the" - I suggest removing the second use of the word "use"

Answer: As per suggestion, we removed the second “use” from the text.

  1. L585 - I suggest removing "crucially" from the sentence, it diminishes the point.

Answer: As per referee suggestions we have removed the word "crucially" in the revised manuscript.

Reviewer 3 Report

I review this manuscript “Prospects of biochar for sustainable agriculture and carbon sequestration: an overview for Eastern Himalayas” and found that it is suitable for the style of review paper. I have an essential comment for authors. This manuscript should define key word “carbon sequestration” because it is similar to “carbon storage”. Whole text is easy to read and the flow is easy to follow. Moreover, I have some special comments as follows.

  1. I have an essential comment for authors. This manuscript should define key word “carbon sequestration” because it is similar to “carbon storage”, such as in Introduction or other chapter. Comparison of these two terms helps audiences understand their difference.
  2. Title and Abstract are suitable. I have no further comment for authors.
  3. In introduction chapter, I only have s slight suggestion in the end of this chapter “This paper is an attempt to highlight the possibilities of biochar production… (lines 87-92)”. It might be considered as “This paper is to present the possibilities of biochar production…”.
  4. I suggest that authors reconsider the chapter titles because it is too much. Such as, “2. Biochar and its Importance”, “3. Properties of Biochar” and “4. Benefits of Biochar” could be combined as one “Importance, Benefits and Benefits of Biochar”, and used sections below chapter to show these chapters.
  5. Lines 566-576, “Future prospects” could be included in “Conclusion”.
  6. Authors could consider that “biochar for sustainable agriculture and carbon sequestration” is global issue or regional issue? If is former, some international papers should be cited to emphasized this key point.

Overall, I am pleased to recommend this manuscript for publication in the sustainability.

Author Response

Comments and Suggestions for Authors

I review this manuscript “Prospects of biochar for sustainable agriculture and carbon sequestration: an overview for Eastern Himalayas” and found that it is suitable for the style of review paper. I have an essential comment for authors. This manuscript should define key word “carbon sequestration” because it is similar to “carbon storage”. Whole text is easy to read and the flow is easy to follow. Moreover, I have some special comments as follows.

  1. I have an essential comment for authors. This manuscript should define key word “carbon sequestration” because it is similar to “carbon storage”, such as in Introduction or another chapter. Comparison of these two terms helps audiences understand their difference.

Answer: We are thankful to the reviewer for this important comment. Carbon sequestration is the process of long-term capturing and storing of carbon dioxide in order to prevent it from entering the atmosphere. Carbon sequestration is the process of removing carbon from the atmosphere and storing it in another form that cannot immediately be released. Carbon sequestration occurs both naturally and as a result of anthropo-genic activities and typically refers to the storage of carbon that has the immediate potential to become carbon dioxide gas

Carbon storage is a complex method which is carried out using various technologies to reduce and capture carbon dioxide and permanently storing them in geological reservoirs. Theoretically, this would prevent those gases from having an effect on climate.

The same has been incorporated in the revised manuscript (Line # 76-84)

  1. Title and Abstract are suitable. I have no further comment for authors.

Answer: We are grateful to the reviewer for the kind remark.

  1. In introduction chapter, I only have slight suggestion in the end of this chapter “This paper is an attempt to highlight the possibilities of biochar production… (lines 87-92)”. It might be considered as “This paper is to present the possibilities of biochar production…”.

Answer: As per referee suggestions we have made the changes in the revised manuscript (Line # 96-98).

  1. I suggest that authors reconsider the chapter titles because it is too much. Such as, “2. Biochar and its Importance”, “3. Properties of Biochar” and “4. Benefits of Biochar” could be combined as one “Importance, Benefits and Benefits of Biochar”, and used sections below chapter to show these chapters.

Answer: As per the suggestion, we have combined the three chapters 2. Biochar and its Importance”, “3. Properties of Biochar” and “4. Benefits of Biochar” into one chapter “2. Importance, Properties and Benefits of Biochar and they have been divided on different sections.

  1. Lines 566-576, “Future prospects” could be included in “Conclusion”.

Answer: As per referee suggestions we have included future prospects within the Conclusion section in the revised manuscript (Line # 583-603)

  1. Authors could consider that “biochar for sustainable agriculture and carbon sequestration” is global issue or regional issue? If is former, some international papers should be cited to emphasized this key point.

Answer: We have considered that “biochar for sustainable agriculture and carbon sequestration” is global issue and accordingly many international papers were cited to justify throughout the manuscript.

Round 2

Reviewer 1 Report

The authors revised the manuscript as per my previous comments.